# Schizophrenic Psychosis Symptoms in a Background of Mild-To-Moderate Carnitine Palmitoyltransferase II Deficiency: A Case Report

**Rochelle N. Wickramasekara** [1] , **Pashayar P. Lookian** [1], **Jeannie Ngo** [2], **Annemarie Shibata** [3] **and Holly A. F. Stessman** [1,*]

[1] Department of Pharmacology & Neuroscience, School of Medicine, Creighton University, Omaha, NE 68178, USA; RochelleWickramasekara@creighton.edu (R.N.W.); PashayarLookian@creighton.edu (P.P.L.)

[2] Department of Psychiatry, School of Medicine, Creighton University, Omaha, NE 68178, USA; JeannieNgo@creighton.edu

[3] Department of Biology, College of Arts and Sciences, Creighton University, Omaha, NE 68178, USA; AnnemarieShibata@creighton.edu

* Correspondence: hollystessman@creighton.edu; Tel.: +1-402-280-2255

**Abstract:** Schizophrenia is a multifaceted mental illness characterized by cognitive and neurobehavioral abnormalities. Carnitine palmitoyltransferase II (CPT II) deficiency is a metabolic disorder resulting in impaired transport of long-chain fatty acids from the cytosol to the mitochondrial inner membrane, where fatty acid β-oxidation takes place. Here, we present an interesting clinical case of an adolescent male that presented with psychosis and a history of mild-to-moderate CPT II deficiency. To identify germline genetic variation that may contribute to the phenotypes observed, we performed whole-exome sequencing on DNA from the proband, unaffected fraternal twin, and biological parents. The proband was identified to be homozygous for the p.Val368Ile and heterozygous for the p.Met647Val variant in *CPT2*. Each of these variants are benign on their own; however, their combined effect is unclear. Further, variation was identified in the dopamine β-hydroxylase (*DBH*) gene (c.339+2T>C), which may contribute to decreased activity of DBH; however, based on the patient's presentation, severe DBH deficiency is unlikely. In conclusion, the variants identified in this study do not clearly explain the observed patient phenotypes, indicating that the complex phenotypes are likely caused by an interplay of genetic and environmental factors that warrant further investigation.

**Keywords:** whole-exome sequencing (WES); carnitine palmitoyltransferase II (CPT II); schizophrenia; psychosis; metabolic deficiency; dopamine-β-hydroxylase; hypoglycemia; seizures

## 1. Introduction

Congenital metabolic diseases are rare inherited genetic disorders associated with a wide range of neurodevelopmental etiologies, including epilepsy, attention deficit disorders, intellectual disabilities, and autism spectrum disorder [1,2]. The vast majority of ATP used by the brain is thought to be supplied by mitochondrial oxidative phosphorylation, not long-chain fatty acid (LCFA) β-oxidation [3]. However, during early postnatal brain development and periods of starvation, neurons use ketone bodies produced by astrocytes or from the liver for β-oxidation and ATP synthesis [4]. β-oxidation of LCFAs depends upon the carnitine transferase system, or "carnitine shuttle", to mediate carnitine-dependent entry of long chain acyl-Coenzyme A (acyl-CoA) into the mitochondrial matrix [5,6] (Figure 1). Acylcarnitine production in the brain is thought to be important for lipid synthesis, stabilization

of membrane composition, modulation of protein synthesis and function, regulation of antioxidant production, and increased cholinergic neurotransmission [7]. The primary carnitine shuttle proteins are mitochondrial carnitine palmitoyltransferase 1 (CPT1) and carnitine palmitoyltransferase 2 (CPT2). In rodent models, *Cpt1a* and *Cpt2* transcript expression is found in almost all central nervous system regions, particularly the brainstem, cerebellum, and spinal cord, with the highest levels being found in the hippocampus and cerebellum [8].

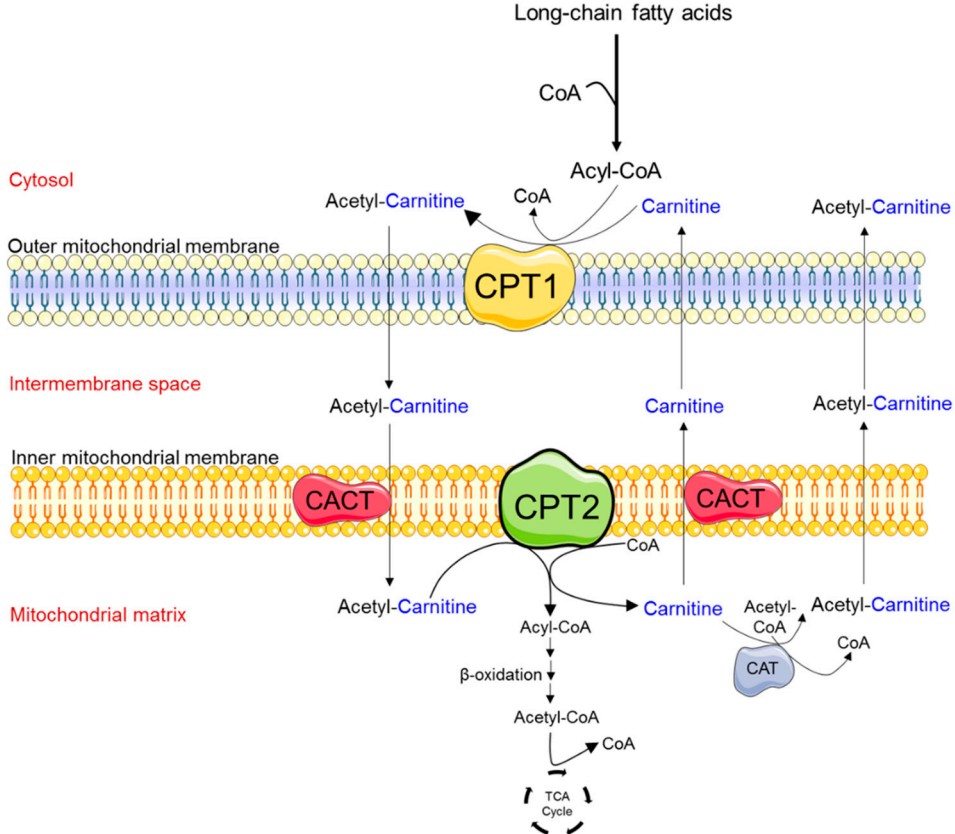

**Figure 1.** Schematic diagram depicting the enzymes involved in the transport of medium- and long-chain fatty acids into mitochondria. A key regulatory step in the process of fatty acid β-oxidation is the entrance of fatty acids into the mitochondria. This process is mediated by the carnitine-shuttle enzyme complex. Fatty acids that enter the cytosol are activated by the enzyme acyl-Coenzyme A (CoA) synthetase to produce long-chain fatty acid acyl-CoAs. Due to the lack of the acyl-CoA transporter in the mitochondrial inner membrane, the acyl group is transferred to the shuttle molecule, carnitine. In a rate-limiting step, carnitine palmitoyl transferase 1 (CPT1) catalyzes the conversion of long-chain acyl-CoAs into acylcarnitine at the mitochondrial outer membrane. Acylcarnitines cross the inner mitochondrial membrane via the carnitine acylcarnitine transporter (CACT). Carnitine palmitoyl transferase 2 (CPT2) is located on the matrix side of the inner mitochondrial membrane, where it transfers back the acyl groups from long-chain acylcarnitines to mitochondrial CoA, forming long-chain acyl-CoAs. Fatty acyl-CoA molecules then undergo the process of fatty acid β-oxidation, producing acetyl-CoA, which enters the tricarboxylic acid (TCA) cycle to produce energy (i.e., Adenosine Triphosphate, ATP). Carnitine can be removed from the mitochondrial matrix by CACT or the enzyme carnitine acetyltransferase (CAT) which can catalyze the transfer of short-chain fatty acids from CoA to carnitine.

Mutations in either carnitine palmitoyltransferase are associated with significant morbidity and mortality. Severe mutations in *CPT1* or *CPT2* (termed CPT I and CPT II deficiencies, respectively) lead to build-up of toxic metabolites, which are fatal shortly after birth. CPT I deficiencies are rare autosomal recessive metabolic disorders that cause recurrent attacks of hypoketotic hypoglycemia, hepatomegaly,

seizures, renal tubular acidosis, and hyperammonemia. CPT II deficiencies are autosomal recessive or compound heterozygous metabolic disorders and are more common than CPT I deficiencies. Clinically, CPT II deficiency may present in different forms: a lethal neonatal form, a severe infantile form, or a mild myopathic form [9,10]. The infantile form occurs within 6–24 months of age and presents with recurrent hypoglycemia, cardiomyopathy, seizures, and multiorgan failure including the respiratory and hepatobiliary systems. For the majority of infantile CPT II deficiency cases, the average prognosis is 12 months with the current gold standard treatment. The myopathic form of CPT II deficiency is most common and manifests from infancy to adulthood. This form is characterized by episodes of rhabdomyolysis triggered by prolonged exercise, fasting, or febrile illness. Treatment for CPT II deficiencies includes avoidance of fasting and intense exercise and a low-fat diet enriched with medium-chain triglycerides and carnitine supplementation [11].

Here, we detail an interesting clinical case of mild-to-moderate carnitine deficiency comorbidity with early childhood seizure, adolescent development of attention deficit hyperactivity disorder (ADHD), and early-onset psychosis consistent with schizophrenia. The coexistence of carnitine deficiency, seizure, and neuropsychiatric disorder suggested the possibility of an ultra-rare genetic variant affecting fatty acid oxidation [12,13]. Whole-exome sequencing (WES) was performed on this proband and his father, mother, and unaffected fraternal twin to identify potential causative variants under a monogenic disease model.

## 2. Case Presentation

### 2.1. Clinical Course

Retrospective chart review, re-contact of this family, and consent for WES was approved by the Creighton University Institutional Review Board (protocol #1172777; approved 11-28-2018) in compliance with all relevant federal, state and local regulations and the Declaration of Helsinki. Informed consent was obtained from the patient, his unaffected twin sister, biological mother and father. The proband (10000.p1) was born demonstrating low APGAR (Appearance, Pulse, Grimace, Activity, and Respiration) scores, fetal distress, and jaundice. Shortly following birth, the proband had a left-sided focal seizure requiring stabilization in the neonatal intensive care unit (NICU). Following this initial event, the remainder of childhood development was uncomplicated until 20–22 months, when the proband was brought to medical attention for a tonic–clonic seizure. Through history provided by the parents, it was determined that the seizure activity was likely precipitated by a hypoglycemic state secondary to poor feeding. Imaging at this time demonstrated no acute findings. However, bilateral adrenal calcifications were incidentally discovered and thought to be due to neonatal hemorrhage. Testing following the episode showed blood glucose levels were down to ~30 mg/dL and insulin levels were essentially undetectable at 3 mlU/L. There was a discrepant elevation in his blood free fatty acid levels at 1.14 mmol/L and low levels of free carnitine, which suggested a defect in fatty acid oxidation. The proband was stabilized by dietary management of carbohydrates, protein, and fats (30% of total daily calories), with feeding intervals every 2–3 h. The remainder of the proband's development was unremarkable, other than a persistent fatigability leading to further medical evaluation at age 11. Metabolic testing for recurrent hypoglycemia showed elevated acylcarnitine levels suggesting either translocase (CACT) or CPT II deficiency [14,15]. Mitochondrial studies from cultured skin fibroblasts were conducted at Baylor College of Medicine on all family members. Probe studies showed an elevation of C16 in cells from the proband (10000.p1), his biological mother (10000.mo) and father (10000.fa), but not his unaffected fraternal twin (10000.s1). CPT2 activity was further tested on the proband with a research protocol [16] in Paris, France, using cultured proband skin fibroblasts. CPT2 activity was examined by measuring the palmitoyl-L-(methyl-14C)carnitine (14C-PalCar) formed from L-(methyl-14C)carnitine and palmitoyl-CoA. Activity levels are reported as nanomole palmitoyl-L-(methyl-14C)carnitine formed per minute per milligram of cell protein (nmol 14C-PalCar/min/mg protein) [17]. Proband CPT2 activity

was reportedly reduced by approximately 50% (0.48) compared to unrelated controls (0.94 ± 0.21; n = 19), suggesting CPT II deficiency. Other biological family members were not tested with this protocol. The patient was diagnosed with mild-to-moderate CPT II deficiency, at which point carnitine supplementation with levocarnitine at the maximum pediatric dose of 3 g/day PO was initiated.

As an adolescent, the proband struggled with concentration, causing a significant impact on his academic performance and social development. After clinical evaluation, the proband was placed on a standard dose of lisdexamfetamine at 40–70 mg/day PO. Around this time, he began to develop persecutory-subtype delusions. The proband reported noticing patterns with significant personal meaning that were demeaning to them. The stimulant was discontinued, but the proband's delusions persisted even after cessation. Stabilization for this acute episode of psychosis was achieved by a multi-drug regimen of risperidone 5 mg/day PO (Per os, oral administration), aripiprazole 2 mg/day PO, mirtazapine 15 mg/day PO, and benztropine 0.5 mg/PO. Following a transition in psychiatric care, an MRI scan showed occipital lobe encephalomalacia suggestive of long-standing cortical injury. The medical intervention was transitioned to the antipsychotic paliperidone at a dosing schedule of 3 mg/day PO. This intervention significantly reduced the proband's psychiatric symptoms and he was transitioned to a long-acting injectable formulation of paliperidone (234 mg every 28 days) for maintenance therapy. Despite sustained improvement with this therapeutic intervention, the proband continues to report sporadic issues with ideas of reference and racing thoughts but has no other functional deficits or significant comorbidities at this time.

### 2.2. Family-Based Sequencing Identifies Compound CPT2 Variation in the Proband

To identify germline genetic variation that may contribute to the phenotypes observed in this proband, we performed WES on saliva DNA [18,19] from the proband, his unaffected fraternal twin, and biological mother and father. This strategy has yielded much fruit under a variety of neurodevelopmental disease models [20–23]. The fraction of exome targets that were covered at ≥20X depth was ≥95% across all family samples. Sequence mapping statistics suggested that all family samples were of similar quality and could be compared (Table S1). Variant calling for single-nucleotide variants (SNVs) and insertions–deletions (INDELs) showed that all family members carry a similar load of genetic variation (Table S2). In the proband, 10,502 coding SNPs (Table S3) and 623 INDELs (Table S4) were identified. Of these variants, 93% were predicted to be missense, 5.5% frameshifting, 0.71% nonsense, and 0.70% splicing. As expected, the vast majority of these variant alleles were inherited (Table S5). For many common heterozygous variants identified in the proband, the mother and father were both carriers. Given our short-read data, phasing of these variants was not possible (i.e., "origin unknown"; Tables S3–S5). For those variants that could be phased, 57% were inherited from the father and 43% from the mother (Tables S3–S5). Two potentially de novo coding variants were identified in the proband—a missense variant (p.Cys56Gly) in the gene *TMEM196* (Table S3) and an in-frame deletion variant (p.Pro12del) in the gene *LONRF2* (Table S4). Literature studies were not available for either of these genes supporting their potential involvement in the phenotypes observed in this proband.

Based on the previous CPT2 functional findings, we initially focused on germline genetic variation in the carnitine synthesis pathway. Variants of interest were validated by Sanger sequencing (primers shown in Table S6). The first of two identified *CPT2* variants (NC_000001.10:g.53676448G>A (GRCh37); NM_000098.2:c.1102G>A; NP_000089.1:p.Val368Ile) were transmitted to the proband (homozygous) from both the mother and father, who are heterozygous carriers. The unaffected twin did not inherit this variant (wild-type; Figure 2). This variant is common among individuals of European ancestry (allele frequency of ~54%; Table S7) and can be found under the following accessions: ClinGen: CA285313; OMIM: 600650.0018; dbSNP: rs1799821. This single base pair variant results in an amino acid change at position 368 from Val to Ile, a conservative change. Multiple in silico algorithms predict this variant to be benign (Table S3), in agreement with the clinical citations in ClinVar. Compound V368I variation with F352C (NG_008035.1:g.19301T>G; NM_000098.2:c.1055T>G; NP_000089.1:p.Phe352Cys;

rs2229291) has been associated with decreased CPT2 activity, stability [24], and influenza-associated encephalopathy (IAE) [25,26]. However, the F352C variant was not identified in any members of this family.

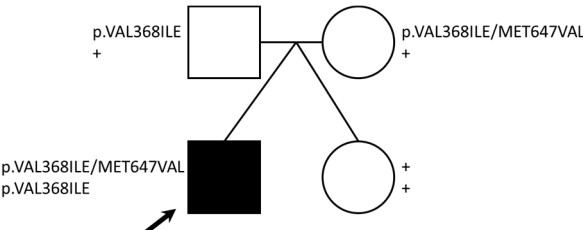

**Figure 2.** *CPT2* variants identified. The pedigree shows genotypes for two *CPT2* missense variants identified for each family member. Note: + = wild-type genotype, black arrow = proband.

A second identified *CPT2* variant (NC_000001.10:g.53679229A>G (GRCh37); NM_000098.2:c.1939A>G; NP_000089.1:p.Met647Val) was transmitted to the proband from the mother. Both the mother and the proband are heterozygous for this variant, whereas the father and twin are wild-type (Figure 2). This variant, although less common than Val368Ile, has a frequency of 13–22% among individuals of European ancestry (Table S8) and can be found under the following accessions: ClinGen: CA285315; dbSNP: rs1799822. This single base pair variant results in an amino acid change at position 647 from Met to Val. While the SIFT algorithm predicts that this variant may disrupt CPT2 protein function (Table S3), clinical annotations predict that this variant is likely benign. The M647V variant has also been identified in several published IAE cases [27].

Assessment for additional variation that may affect carnitine metabolism in this proband revealed one benign heterozygous missense variant in *CPT1* [28] but no variation in the transporter, *SLC25A20* (*CACT*). Taken together, none of the variation identified in the carnitine synthesis pathway could clearly explain the phenotypes observed in the proband. Although the p.Val368Ile and p.Met647Val variants alone in their heterozygote state are predicted to be benign, the effect of the combined homozygous p.Val368Ile and heterozygous p.Met647Val variant, as seen in the proband, is unknown.

### 2.3. Identification of DBH Compound Heterozygous Variants

In order to identify additional germline variation that might be linked to the etiologies observed (specifically infantile hypoglycemia or seizures), we utilized the KEGG DISEASE database [29] to extract all genes that have been associated with these disease terms. We intersected this gene list with the variants identified in the proband (Tables S3 and S4) and identified a variant in dopamine beta-hydroxylase (*DBH*) (NC_000009.11:g.136501834T>C (GRCh37); NM_000787.3:c.339+2T>C; p?) that was transmitted to the proband (heterozygous) from the father, who is also a heterozygous carrier. The mother and unaffected twin were both wild-type for this variant (Figure 3). This variant (also referred to in the literature as IVS1+2T>C) is rare among individuals of European ancestry (allele frequency of 0.087%; Table S9) and can be found under the following accessions: ClinGen: CA339912; OMIM: 609312.0002; dbSNP: rs74853476. This variant results in a canonical splice donor site change in intron 1 of *DBH*, which introduces an early stop codon causing an overall reduction in the amount of functional enzyme made [30,31]. This variant is reported at an increased frequency across DBH deficiency cases in either the homozygous or compound heterozygous state [30–35].

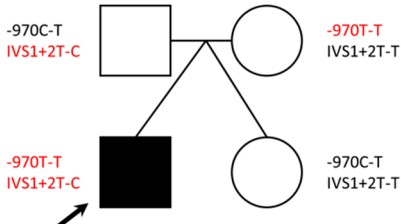

**Figure 3.** *DBH* variants identified. The pedigree shows genotypes for two *DBH* variants identified for each family member. Variants in red have been shown in the literature to have reduced DBH activity. Note: black arrow = proband.

Based on the *DBH*: IVS1+2T>C finding, we performed a literature search for additional *DBH* variation that might affect enzyme activity. In 2001, Zabetian et al. found that a C-to-T polymorphism at nucleotide –1021 in the 5-prime region of the *DBH* gene was related to DBH activity levels in plasma. In a study of 174 European Americans, 16 of TT genotype had DBH activity of 4.1; 46 of CT genotype had DBH activity of 25.2; and 112 of CC genotype had DBH activity of 48.1 nmol/min/mL [36]. The *DBH*-970C-T variant (NC_000009.11:g.136500515T>C (GRCh37)), found under accessions: ClinGen: CA347658 and dbSNP: rs1611115, is present in approximately 22% of individuals of European ancestry [36]. We performed Sanger sequencing on this region in our family samples, as it was not covered by the WES library capture design. The proband and mother both carry TT genotypes (low activity [36]), as compared to the father and the unaffected twin, who are heterozygous (CT genotypes; moderate activity [36]; Figure 3). Taken together, the presence of these two variants in the proband may suggest decreased DBH activity in the body.

DBH is responsible for the conversion of the catecholamine dopamine to norepinephrine, which is required for the proper maintenance of blood pressure and muscle tone. DBH deficiency often presents as hypotension, muscle hypotonia, hypothermia, and hypoglycemia in infancy, progressing to orthostatic hypotension and exercise fatigability in adolescence [37]. While the observed failure to thrive in the proband may be consistent with some aspects of DBH deficiency presentation, assessment for orthostatic hypotension following these genetic findings were negative. A history of hobby distance running in the proband also suggested that severe DBH was unlikely. Therefore, further catecholamine testing was declined.

### 2.4. Polygenic Risk Likely Underlies Schizophrenia Diagnosis in This Case

While the focus of our variant search initially surrounded the early seizure phenotype in the proband, the schizophrenia diagnosis in adolescence could not be ignored. As before, we utilized public databases (ClinVar and SZDB2.0) [38] and publications [39] to curate a list of genes that have been associated with schizophrenia risk and intersected this list with our variant dataset (Tables S3 and S4). Importantly, we did not identify any protein-coding variation in the proband in genes that have been associated with rare de novo mutations in the most current large schizophrenia sequencing datasets [39] (Table S3). Further, the number of coding autosomal variants (missense, nonsense, and splice only) in genes associated with schizophrenia pathogenicity in ClinVar or previous genome-wide association studies (GWAS) in the proband was not significantly different than his unaffected twin (Fisher's exact test; $p > 0.68$). While particular variant combinations (polygenic risk) versus sheer number of variants carried may be more important for pathogenicity in this case, we were underpowered to assess this in this study.

### 2.5. Incidental Findings

Of note, we identified a variant in the gene *MUTYH* (NC_000001.10:g.45798475T>C (GRCh37); NM_001048171.1:c.494A>G; NP_001041636.1:p.Tyr165Cys) that can be found under the following accessions: ClinGen: CA011761; OMIM: 604933.0001; dbSNP: rs34612342 in this family. This single-base

pair variant results in an amino acid change in the protein at position 165 from Thr to Cys. Both 10000.fa and 10000.p1 are heterozygous carriers of this variant. Overall, the lifetime risk of developing colorectal cancer is estimated at 1 in 323 (4.4%) for men and 1 in 525 (4.1%) for women [40]. This variant has been found in individuals with colorectal cancer and is characterized as a risk variant, but is considered to be low penetrance [41].

## 3. Discussion

The proband in this study was considered for WES due to a complex combination of metabolic and neuropsychiatric phenotypes. What is unknown from the medical record is whether the proband suffers from two independent clinical phenotypes—seizures caused by hypoglycemia and schizophrenia—or if all phenotypes are a complex presentation resulting from a rare de novo genetic variant not present in other immediate family members. The availability and willingness of the biological mother, father, and a sibling to also be sequenced increased our chances of identifying causative variation in the proband, as evidenced by previous studies [42,43]. As twin studies of neuropsychiatric disorders point to some combination of both genetic and environmental contributions to disease presentation, the availability of a fraternal twin sibling offers the opportunity to reduce environmental contributions. However, this study design likely carries no greater power than any two siblings raised together as the twins are of different sexes and one of the phenotypes in this case is schizophrenia, which carries a significant sex bias.

Whole-exome sequencing data were used to test the hypothesis that a rare de novo germline variant is present in the proband, that uniquely links the multiple phenotypes. While rare monogenic-like disorders are thought to represent only a very small fraction of schizophrenia cases [39], a monogenic etiology could not be ruled out given the phenotypes observed in this proband and the substantial evidence supporting a clinical correlation between epileptiform seizure disorders and schizophrenia-like psychosis [44–46]. Indeed, clinical outcomes in patients with a rare monogenic epilepsy disorder resulting from *KCNQ2* mutations [47,48] ranged from benign familial neonatal–infantile seizures (BFNIS) to severe neonatal-onset epileptic encephalopathy (EE) [44]. Previous studies have established that the risk of schizophrenia-like psychosis is 6–12 times more likely in patients with epileptic seizures than in the general population [46,49]. Recognizing the significant impact of psychiatric comorbidity in this patient population, we proceeded to evaluate this hypothesis. We identified one de novo coding SNV in the proband—a missense variant in *TMEM196* (p.Cys56Gly). Sequencing of over 60,000 control humans suggests that *TMEM196* is mutable (pLI = 0); however, this specific variant (p.Cys56Gly) has not been previously observed [50] and the Cys56 amino acid is highly conserved across many species. While this variant is predicted to be disrupted by SIFT [51] and PolyPhen [52] algorithms, little is known regarding the TMEM196 protein's function. Extensive functional work would need to be done to link this variant to the phenotypes observed in this proband. Additional de novo INDELs were identified but none could be verified by Sanger sequencing.

Carrying specific combinations of common variation has been significantly associated with schizophrenia risk; this can be calculated using SNP arrays and is termed a polygenic risk score (PRS). PRSs consider many hundreds of thousands of common SNPs spanning the genome (including much of the non-coding space) to link loci to phenotypes. Unfortunately, most of these SNPs cannot be extracted from WES data, as WES captures only the coding regions of the genome. For example, given the current state-of-the-art Infinium™ PsychArray-24 v1.3 BeadChip, only 289,932 out of a total 593,836 (48.8%) targets were covered in our WES capture design. While a PRS analysis of this family would certainly be interesting, it is beyond the scope of this study (variants identified in the proband (Table S3 and S4) and sibling that have been noted as significant GWAS-tagged genes are highlighted in Table S10). We did, however, look for combinations of variants in the proband that have been linked to either seizures or schizophrenia in the literature. We manually curated lists of genes linked in the literature to the following terms: hypoglycemia, seizure, epilepsy, schizophrenia, psychosis, and carnitine deficiency. While CPT II deficiency was strongly suspected, our results did not reveal any clear pathogenic *CPT2*

variation. What is not known is how multiple common variants may work together in either an additive or synergistic way to create a deficiency (seen in the proband) that is not seen with any one variant alone (as in the case of both parents in this study). Indeed, the two variants identified in our proband, Val368Ile and Met647Val, have been suggested by others as potential susceptibility variants in CPT II deficiency [53]. This is further supported in the case of our proband by the fact that carnitine supplementation plus meal control appeared to have a positive impact. Schizophrenia has previously been linked to abnormal lipid metabolism [45,54,55]. Serum long–chain fatty acids along with β-oxidation are significantly increased in schizophrenia patients compared to healthy controls, presumably the result of insufficient brain energy supply [55]. Increased carnitine transfer and upregulation of transcripts related to carnitine shuttling is observed as a counter mechanism in the brain during periods of starvation [54]. In line with this, prefrontal cortical tissue from schizophrenia patients show increased *Cpt1* and *Cpt2* transcripts [54]. However, *CPT2* variation and CPT deficiencies have not been directly linked to schizophrenia diagnosis, suggesting that additional genetic variation may be at play with the adolescent neuropsychiatric phenotypes observed. It is likely that dysregulation of fatty acid β-oxidation and the brain energy supply is related to the disease and may be amplified during critical periods of brain development; however, this requires further investigation.

We also identified likely disruptive variation in the *DBH* gene, which would be predicted to result in increased levels of dopamine in the body. Interestingly, dopamine has been the primary catecholamine target of most schizophrenia research and drug development to date, as early studies of the disorder suggested that excess dopamine may be causative [56]. While DBH deficiency has not been directly linked to schizophrenia, it is possible that the *DBH* variation in this proband contributes to his psychosis. This will require additional research. However, it is clear that the *DBH* variants identified in this proband do not result in classical DBH deficiency syndrome [31].

In conclusion, while WES sequencing in this family did not identify any clear pathogenic variants related to the medical management of the proband's clinical phenotype, it did provide new hypotheses for testing in the laboratory. It is most likely that this case represents multiple clinical conditions underscored by a combination of genetic variants inherited from both parents. WES also provided incidental findings that may be of clinical use to the family in the future, e.g., the *MUTYH* variant identified in the father and proband and a Leiden factor V (*F5*) variant identified in the mother and twin sister (not shown), which increase the risk for developing blood clots when taking estrogen-containing contraceptive pills.

**Supplementary Materials:** The following are available online at http://www.mdpi.com/2571-841X/3/4/31/s1: Table S1: Sequencing metrics for family 10000. Table S2: Variant summary for family 10000. Table S3: Coding SNPs identified in the proband. Table S4: Coding INDELs identified in the proband. Table S5: Inheritance of coding variants in the proband. Table S6: Sanger primers for orthogonal validation. Table S7: Population frequencies of rs1799821 from the ExAC database. Table S8: Population frequencies of rs1799822 from the ExAC database. Table S9: Population frequencies of rs74853476 from the ExAC database. Table S10: Coding SNPs identified in the sibling.

**Author Contributions:** Conceptualization, H.A.F.S.; methodology, H.A.F.S.; validation, R.N.W., H.A.F.S.; formal analysis, H.A.F.S.; resources, H.A.F.S., A.S.; writing—original draft preparation, R.N.W., P.P.L., J.N.; writing—review and editing, H.A.F.S., A.S.; supervision, H.A.F.S.; funding acquisition, H.A.F.S., A.S. All authors have read and agreed to the published version of the manuscript.

**Funding:** This work was funded by the LB692 Nebraska Tobacco Settlement Biomedical Research Development Program to H.A.F.S. and the Health Science Strategic Investment Fund (HSSIF) to A.S.

**Acknowledgments:** We would like to thank the participants and clinician who participated in this study. Sequencing, alignment, and variant calling were performed at the Beijing Genomics Institute (BGI) (Hong Kong).

**Conflicts of Interest:** The authors declare no conflict of interest.

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
