# Peer review of "Schizophrenic Psychosis Symptoms in a Background of Mild-To-Moderate Carnitine Palmitoyltransferase II Deficiency: A Case Report"

_reports, doi:10.3390/reports3040031_

Round 1
Reviewer 1 Report
The authors, organized from researchers at Creighton University,
studied and described the genetic background of a patient with schizophrenia related psychotic symptoms (episodes) and the proband's family members
in their article. The genetic pattern of the proband, as well as his twin sister and both parents, was examined using the whole exome sequencing (WES) method. Several important mutant loci were identified in the family tree analysis, of which genetic variants related to carnitine palmitoyltransferase II (CPT II) enzyme deficiency and dopamine β-hydroxylase (DBH) gene variants were analyzed in detail. The obtained sequencing data were used to test two different hypotheses, (1) a de novo germline variant is present that uniquely links the multiple phenotypes connected to the probandand; and (2) independent sets of variants are linked to independent phenotypes. The authors concluded that the genetic alterations (homo- or heterozygous mutations) identified in this study do not clearly explain the observed patient phenotypes, however, the results of the present study and especially further regular genetic analyzes may help to scientifically explore the possible causes and etiology of schizophrenia.
I find the presented scientific material worth publishing, therefore I support its appearance.
Author Response
We thank the reviewer for their comments.
Reviewer 2 Report
Review report
Wickramasekara and colleagues describes a whole-exome sequence study of a small family with the proband having psychosis and history of mild CPT-II deficiency. They identified two (benign) missense variants segregating in the family, but these variants could not explain the phenotypes of the proband.
General comments
1: I am not too familiar with the terminology in case reports, but I find it very confusing to use “their”, “them” and “they” when talking about the proband. Clearly this is grammatically incorrect, but it could be the terminology used in case reports.
2: I really lack a clear description of the sequencing protocol. I Find the supplementary methods a bit confusing to read. For example, page 2 first paragraph. “In addition, the strict quality control…” What are the QC filters? Is that what is described in third paragraph on page 2? If you, it would have been clearer if that came after mentioning the QC. Also, what reference genome were the reads mapped to? What does “perform a series of annotation for variants” mean? You have a set of variants which you simple annotate – what does “series” mean? I think the methods could be easier to read if you for example change the font of the code chunks, so it is easier to see what is text and what is code.
3: Page 5 section 2.3 you write you “further mined the WES data for additional germline variation that might be linked to the etiologies observed”. Please described how this was done – it must involve some data analysis?
4: You say you identify “germline genetic variation” (page 4, line 136). How can you do that from saliva? In that case it must be somatic mutations?
5: I lack a better description of all the variants identified. There must be more variation between the four individuals than what is described here. How many segregating variants were found in total; how many were missense, loss of function, stop gained etc.
6: I find it a bit difficult to understand that the authors seek to find “one or several genetic variants” (p 6, line 222-233) explaining the phenotype since it is widely known (and accepted) that schizophrenia is a strongly polygenic disease. Please provide a better explanation for this statement.
7: Your second hypothesis is: “independent sets of variants are linked to independent phenotypes”. This would they imply that the proband has two monogenic diseases which we know is not the case (at least for schizophrenia).
8: Many very large genetic analyses have been conducted on schizophrenia. Does the proband has an accumulation of risk variants compared to the three other family members?
Round 2
Reviewer 2 Report
The manuscript has been substantially improved, and I find it ready for publication.